# Effects of *Sphingobium yanoikuyae* SJTF8 on Rice (*Oryza sativa*) Seed Germination and Root Development

Ying-Tzy Jou [1] , Elmi Junita Tarigan [2,3], Cahyo Prayogo [3], Chesly Kit Kobua [4], Yu-Ting Weng [1] and Yu-Min Wang [5,*]

[1] Department of Biological Science and Technology, National Pingtung University of Science and Technology, Pingtung County 91201, Taiwan

[2] International Master Program in Soil and Water Engineering, National Pingtung University of Science and Technology, Pingtung County 91201, Taiwan

[3] Department of Soil Science, Faculty of Agriculture, Brawijaya University, Jl. Veteran 1, Malang 65145, Indonesia

[4] Department of Tropical Agriculture and International Cooperation, National Pingtung University of Science and Technology, Pingtung County 91201, Taiwan

[5] General Research Center, National Pingtung University of Science and Technology, Pingtung County 91201, Taiwan

[*] Correspondence: wangym@mail.npust.edu.tw; Tel.: +886-8-770-3202 (ext. 6394); Fax: +886-8-774-0220

**Abstract:** The interaction between plant roots and rhizobacterium communities plays a crucial role in sustainable agriculture. We aimed to assess the effects of *Sphingobium yanoikuyae* SJTF8 on rice seed germination and development, as well as to observe the effects of different concentrations of *S. yanoikuyae* SJTF8 on the root systems of rice seedlings. The bacteria are best known for their role in the bioremediation and biodegradation of pollutants, and thus far, there is research that supports their agricultural prospects. The experiment comprised five different *S. yanoikuyae* SJTF8 concentrations: SP-y 8 ($10^8$ CFU/mL); SP-y 7 ($10^7$ CFU/mL); SP-y 6 ($10^6$ CFU/mL); SP-y 5 ($10^5$ CFU/mL); SP-y 4 ($10^4$ CFU/mL). We used sterilized water as the control treatment. The bacteria triggered the synthesis of IAA, while the seedling root lengths substantially increased on the 12th day after germination. The high application concentrations of *S. yanoikuyae* SJTF8 resulted in higher IAA production (with the SP-y 7 and SP-y 8 concentrations ranging from 151,029 pg/mL to 168,033 pg/mL). We found that the appropriate concentrations of *S. yanoikuyae* SJTF8 when applied as an inoculant were SP-y 7 and SP-y 6, based on the increased root growth and biomass production. The bacteria were also able to solubilize phosphorous. The growth response from the rice seedlings when inoculated with *S. yanoikuyae* SJTF8 presents the potential of the bacteria as a growth promotor. Its application in rice cultivation could be a sustainable approach to rice production.

**Keywords:** bio-priming; seed germination; plant-growth-promoting rhizobacteria; *Sphingobium*; phytohormones; indole-3-acetic acid; phosphorus solubilization; root system; rice

## 1. Introduction

Land degradation is a critical issue in agriculture that contributes to poor production [1,2]. Researchers have shown that the loss of biodiversity and the ecosystem is due to human activities [1,3]. One factor that causes land degradation is the intensive use of chemical fertilizers for agricultural purposes [1,4,5], which. Therefore, we need sustainable agricultural development to provide adequate yields without causing further harm to the soil and environment [6,7]. Scholars have proposed this not only for food production, but also to enhance sustainability [8–10].

An effective strategy to reduce the usage of chemical fertilizers is through the application of beneficial bacteria. These organisms are known to thrive in a wide range of environments. Rhizospheric bacteria have a particular importance to plant life. In early

studies, researchers found that bacterial communities competitively colonize the roots of plants, which results in various interactions between the host plants and these bacterial communities that may stimulate growth or reduce pathogenic infection. Kloepper and Schroth refer to these beneficial bacterial species as plant-growth-promoting rhizobacteria (PGPR) [11]. According to Reddy [12], PGPR are an indispensable portion of the rhizosphere biota that are grown in association with the host plants and have the potential to stimulate their growth. Some of the known PGPR strains that researchers have identified in the genera are as follows: *Agrobacterium*; *Arthrobacter*; *Azotobacter*; *Azospirillum*; *Bacillus*; *Burkholderia*; *Caulobacter*; *Chromobacterium*; *Enterobacter*; *Erwinia*; *Flavobacterium*; *Micrococcus*; *Pseudomonas*; *Serratia*; *Stenotrophomonas*; *Trichoderma* [12–15]. These PGPR have various degrees of effects that trigger direct or indirect mechanisms in plants, which include the mobilization of soil nutrients (such as phosphorus, potassium, zinc, iron, and other essential mineral nutrients), the synthesis of phytohormones in plants (such as auxins, cytokinins, and gibberellins), the amelioration of the soil structure, and bioremediation [16–21]. Through parasitism, PGPR inhibit the occurrence of phytopathogens and other damaging microbes. They prompt the production of antagonists and microbial products, as well as promote systemic stress tolerance in host plants [22,23]. Understanding the beneficial effects of PGPR provide opportunities if utilized, and they may be a sustainable approach to food crop production.

A less known bacteria species that is utilized in agriculture is the *Sphingobium yanoikuyae* SJTF8, which is from the genera *Sphingobium* under the class *Alphaproteobacteria*. The genus is well known for its role in the bioremediation and biodegradation of pollutants in sediments and sandy soils [24–27]. Various *Sphingobium* species stimulate the production of phytohormones, such as salicylic acid (SA), indole-3-acetic acid (IAA), and Zeatin abscisic acid (ABA), in addition to other essential biosyntheses in host plants, corroborating the genera's potential to be used as PGPR [28–32]. *S. yanoikuyae* itself also has a similar effect on plant life. Researchers widely examine it for its ability to degrade harmful substances that have detrimental effects on human health [27,29,33–35]. The bacteria species also triggered an increase in the root growth and overall plant biomass production [29]. Hence, the researchers considered it to function as PGPR. Rincón-Molina et al. [36] inoculated pepper (*Capsicum chinense*) with *S. yanoikuyae* NFB69 and observed substantial increases in the crop's plant height, root length, and stem diameter. Furthermore, the bacteria greatly improved the crop's biomass and increased the number of fruits produced. Other researchers observed a similar response when they applied the bacteria (*S. yanoikuyae*) to rice (*Oryza sativa*) under moisture deficit conditions [37]. Its combination with a *Bacillus* spp. and *Burkholderia* spp. reduced the demand for synthetic fertilizers in rice by 25–50%. According to the results, the combination of the three bacteria triggered a substantial increase in the crop's biomass, which eventually improved the grain yield production by 14%. The combination of these bacteria species also influenced the soil nitrogen (N) availability [38]. Both Chen et al. [29] and Poonguzhali et al. [39] reported that *S. yanoikuyae* stimulated the production of IAA in rootlets, which indicates that *S. yanoikuyae* was able to promote the growth of its host plant.

PGPR application, whether applied to the soil or plant foliage or by seed inoculation, plays a role in the cycling of nutrients, providing advantages to plants [22]. Their presence in the host plant's root system has been shown to increase the lateral root length and stimulate the root hair elongation, increasing the water and nutrient uptake and plant advancement [23]. Some plants have distinct hormone-distribution and production patterns. As a result, understanding the interactions between plants and microorganisms that can boost plant growth is critical. In this study, we aimed to evaluate the effect of *S. yanoikuyae* SJTF8 on the rice seed germinating efficiency and root development. Furthermore, we aimed to observe the effects of different concentrations of *S. yanoikuyae* SJTF8 on the rootlets of rice seedlings. The information generated from this study may be applied in rice cultivation and could support a sustainable approach to rice production.

## 2. Materials and Methods

### 2.1. Site Description

We conducted this study at the Department of Biological Science and Technology, the National Pingtung University of Science and Technology (NPUST). The crop of interest was rice (*Oryza sativa*). Thus, we used a high-yielding aromatic rice variety designated as Kaohsiung 147 (KH-147). We obtained the rice seeds from the Department of Food Science at NPUST.

### 2.2. Isolation Bacteria

We collected the bacterial species in a tropical rainforest area in Lundu, Sarawak, Malaysia (GPS location: 1°41′26.0″ N and 109°50′44.9″ E). We sampled the bacterial samples using a microbiological air sampler apparatus (Coriolis Micro M. Bertin Technologies). We randomly placed the device at five locations within the vicinity of the rainforest area. At each site, we placed the device on the soil surface with the air-suction tube leveled approximately 20 cm off the ground (Figure 1). We maintained all the bacterial isolates on a starvation medium nutrient containing lysogeny broth (LB). The agar medium consisted of 2.5 g L$^{-1}$ of glucose (4.8 g L$^{-1}$), lactose (2.5 g L$^{-1}$), and sodium acetate (20 g L$^{-1}$). We left the inoculated media at room temperature for five days. We repeated the procedure three times to obtain pure colonies. We sub-cultured the samples once a month for the duration of the study, and we stored them at −80 °C.

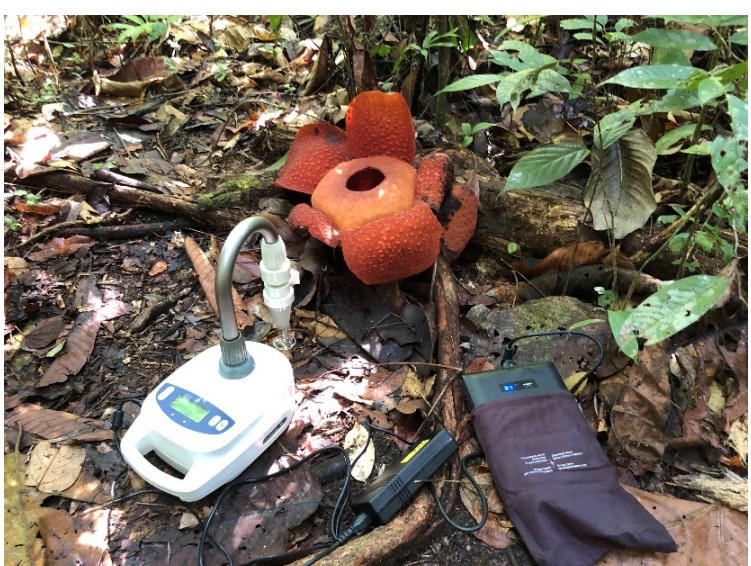

**Figure 1.** Air sampling device used in rainforest of Sarawak, Malaysia.

### 2.3. Strain Identification

We sent the samples to the Food Industry Research and Development Institute, Hsinchu, Taiwan (ROC), for identification. The researchers performed the bacterial DNA isolation and identification using matrix-assisted laser desorption/ionization time-of-flight (MALDI-TOF) mass spectrometry, and they used the MALDI Biotyper Repository for the sequence interpretation. The PCR mixture of 30.0 μL consisted of 10 μL PCR buffer, 10 nM dNTP, and 0.3 μL template DNA. The amplification was carried out in a thermocycler at 94 °C for 5 minutes (min), followed by 40 cycles of 30 seconds (s) at 95 °C, 30 s at 55 °C, and 2 min 20 s at 72 °C, with a final extension at 72 °C for 5 min. The researchers amplified the 16S rRNA gene using the primers 16S F1 (AGAGTTT-GATCATGGCTCAG) and 16S R1 (GGCTACCTTGTTACGACTT), which were complementary to the 5′ and 3′ ends of the prokaryotic 16S rRNA, respectively. We exported the rRNA gene sequence to the basic local alignment search tool (BLAST), and we confirmed the bacterium against the National Center for Biotechnology Information (NCBI) database. We initially isolated nine bacteria

species from the air sample. From the nine species, we did not consider those with partial sequences or unspecified descriptions for the study (Table 1).

**Table 1.** Details of bacteria species used in study.

| Entry | Closely Related Taxa | Strain Type (Gene Bank ID) | Similarity |
|---|---|---|---|
| SP-y | *Sphingobium yanoikuyae* | SJTF8 | 100.00% |
| - | *Sphingobium yanoikuyae* * | HAMBI 1842 | 100.00% |
| - | *Sphingomonas* sp. * | clone 1 | 100.00% |
| - | *Sphingomonas* sp. * | SaMR12 | 100.00% |
| - | *Sphingomonas* sp. * | P2 | 100.00% |
| - | *Sphingomonas paucimobilis* * | ZFJ-16 | 99.93% |
| - | *Bacterium* * | M24(2011) | 99.93% |
| - | Uncultured bacterium * | clone5'-60 | 99.93% |
| - | Uncultured bacterium * | EDW07B005_154 | 99.93% |

Source: DNAeasy Plant Kit (Mission Biotech, Taiwan). The symbol * indicates that the strain was partially sequenced.

### 2.4. Hemolysis Test

We also conducted a hemolysis test on the blood agar. By applying the streaking method, we cultured a pure strain of the *S. yanoikuyae* SJTF8 onto the blood agar media before placing it in an incubator for 72 h (h) at 37 °C. We performed the visual assessment guided by the three hemolysis test references: alpha ($\alpha$) hemolysis, beta ($\beta$) hemolysis, and gamma ($\gamma$) hemolysis.

### 2.5. Phosphate Solubilization Test

We assessed the bacteria's ability to solubilize phosphorus (P) to phosphate ($PO_4^{3-}$) by culturing the pure strain of *S. yanoikuyae* SJTF8 in Pikovskaya agar medium at room temperature for 72 h. We used a qualitative approach of assessment to assess the bacteria's ability to solubilize P. We performed the inoculation period observations by identifying the halo zones around the *S. yanoikuyae* bacterial colonies that grew on the Pikovskaya agar.

### 2.6. Seed Sterilization

We achieved the surface sterilization of the rice seeds by treating them with 10 mL of sodium hypochlorite (NaOCl) for 10 min. We washed the seeds three times using sterile distilled water, and we air-dried them before the inoculation.

### 2.7. Bacterial Concentration

We cultured the purified colony of *S. yanoikuyae* in 1 L of LB liquid medium at 25 °C for 24–36 h. We then measured the bacterial suspension with a spectrophotometer at OD600 to ensure that this value was at 1 ($2 \times 10^9$ CFU/mL = 1.0) [36]. We determined the five different CFU concentration levels through the process of serial dilution. In descending order, the concentration levels were as follows: $10^8$, $10^7$, $10^6$, $10^5$, and $10^4$ CFU/mL. For the trial, we used a randomized complete block design (RCBD) comprising four replicates to evaluate the effects of the different concentrations on the assessment parameters.

### 2.8. Seed Germination

We used six treatments in this study. We labeled the five concentrations as follows: Sp-y 8 ($10^8$ CFU/mL); SP-y 7 ($10^7$ CFU/mL); SP-y 6 ($10^6$ CFU/mL); SP-y 5 ($10^5$ CFU/mL); SP-y 4 ($10^4$ CFU/mL). We assigned sterilized water as the control treatment. We assigned each treatment four replicates each. Each replicate per treatment consisted of 10 rice seeds of KH-147. We soaked the sterilized rice seeds for one hour in the respective tubes containing the six treatments. We repeated this with the control treatment, in which we soaked the sterilized rice seeds in sterilized water for one hour. Following inoculation, we aseptically bedded the seeds onto soaked filter papers inside sterilized containers. We stored the

containers in a dark cabinet at $25 \pm 2$ °C for 72 h before removing them and exposing them to light. We regularly added distilled water to prevent the filter paper from drying.

### 2.9. Indole-3-Acetic Acid Production

We performed the quantitative analysis of the indole-3-acetic acid (IAA) using ELISA Kits (Catalog No. RK00676, AB Clonal Technology, Boston, Maryland, MA, USA). We sampled the rice seedlings after 12 days of germination. We dissected the seedlings, and we immersed the roots of the rice seedlings per treatment in liquid nitrogen before grinding them. We added about 1 g of ground plant tissues to 6 mL of 80% methyl alcohol and allowed it to rest on a vibrator for 24 h at 4 °C. We centrifuged the mixture of 2.0 g at 4 °C for 10 min to separate the methyl alcohol from the supernatant. After pipetting approximately 2 mL of the methyl solution to a new test tube, we added the remaining supernatant with 2 mL of 80% alcohol and vibrated it for another 1 h at 4 °C. We then centrifuged the sample, and we extracted the remaining methyl solution and added it to the initial methyl solution, which brought the total solution to 4–5 mL. We then evaporated the methyl solution of the respective treatments to about 2 mL using a vacuum-drying evaporator before we added 1 mL of petroleum ether. After we separated the two solutions, we removed the layer of petroleum ether, and the methyl alcohol was ready for use. We prepared the standards and reagents by adding 50 μL of the standard and test samples to each well, followed by the addition of 50 μL Biotin Conjugate Antigen Working Solution and an incubation period of 1 h at 37 °C. After rinsing, we added 100 μL of Streptavidin-HRP Working Solution and incubated the sample for 30 min at 37 °C. We applied an additional 90 μl of Substrate Solution and again incubated the samples for another 15–20 min at 37 °C in a dark cabinet. We applied 50 μL of Stop Solutions to the samples before observing them under a photospectrometer. We set the measuring absorbance wavelength at 450 nm, and we set the correction wavelength at 570 nm or 630 nm.

### 2.10. Plant Growth Parameters

We performed the plant growth observations from the 1st to 14th day after the seedling germination. We measured the seedling root and shoot lengths using a ruler. We used ImageJ software (Java 8: U. S. National Institutes of Health, Bethesda, Maryland, MA, USA) to observe the root air and presence of bacteria in and around the seedling rootlets. We calculated the germination rate, or seedling emergence, speed of germination, and seedling vigor index using the following formulas of Gholamalizadeh et al. [40].

We calculated the percentage of germinated seed as follows:

$$Seed\ germination\ percentage = \frac{Number\ of\ seeds\ that\ germinate}{Number\ of\ seeds\ sown} \times 100\% \tag{1}$$

We measured the rate of seed germination as follows:

$$Germination\ speed = \frac{Number\ of\ seeds\ germinated\ on\ the\ day}{Days\ after\ sowing} \tag{2}$$

We determined the seed vigor index as follows:

$$Seed\ vigor\ index = Germination\ percentage \times seedling\ lenght \tag{3}$$

We assessed the seed germination parameters over a period of 7 days. We assessed the percentage of the seed germination on the fourth day after exposing the seeds to their respective treatments. We observed the speed of germination daily for 5 days. We assessed the seed vigor index on the sixth day after exposing the seeds to their respective treatments.

### 2.11. Observations under Microscope

We observed the root hairs and presence of bacterial colonization in the roots using Olympus DP73-digital microimaging and Olympus cellSens software (version 1.5, Amer-

ican Laboratory, San Francisco, CA, USA). To acquire the images, we dissected random samples of the seedling roots to small sizes and successively immersed them in Baso Rapid Gram Stain (BaSO Biotech Co. LTD, Taiwan, R.O.C) for 30 s in each solution. We repeated this process for all six treatments.

### 2.12. Statistical Analysis

We tabulated all the raw data using 2019 MS Excel. We used the IBM SPSS program to perform the statistical analysis on the data (version 20, IBM, Armonk, New York, NY, USA). We conducted an analysis of variance (ANOVA). We compared the treatment means using Duncan's multiple range test (DMRT) at a 5% significance level.

## 3. Results and Discussion

### 3.1. Strain Identification

We performed the strain identification using the 16S RNA sequencing method, in accordance with Johnson et al. [41]. We isolated nine bacterial strains. The bacteria strain used in this study was the *S. yanoikuyae* strain SJTF8 (Table 1). According to the laboratory analysis, the strain contained a complete genome. We did not use strains that had partial genomes or unspecified descriptions in this study.

The *Sphingobium* genus is classified as an *Alphaproteobacteria*, and it is strictly aerobic. Researchers first discovered it in Papua New Guinea (PNG) in 1990. The bacteria is a short rod-shaped Gram-negative nonmotile non-spore-forming chemoheterotrophic species that is yellow or off-white [24,42]. However, due to the genus hosting more than 20 species with diverse properties, researchers have further subdivided it into four different genera: *Sphingobium*, *Novosphingobium*, *Sphingosinicella*, and *Sphingopyxis*. The genus consists of several species that are well known for their roles in the bioremediation and biodegradation of pollutants [25–27,29].

### 3.2. Hemolysis Test

The use of beneficial microorganisms is an attractive option to mitigate the damages that are caused by conventional farming. However, the biosafety status of these proposed microorganisms must be ensured to avoid any risk to the health of humans, animals, or plants [43,44]. Thus, we selected blood agar due to its simplicity, quick implementation, affordability, and compatibility with common laboratory equipment. It has been widely used for clinical, industrial, and agricultural purposes [45–47].

According to Savardi et al. [48], there are three types of hemolysis: alpha (α) hemolysis, beta (β) hemolysis, and gamma (γ) hemolysis. We identified the reaction of *S. yanoikuyae* SJTF8 bacteria on the blood agar as gamma hemolysis. We identified the gamma (γ) hemolysis by the absence of hemolysis, whereby no blood agar was absorbed by the bacteria (Figure 2). The bacterium did not break down the blood cells, which indicates that it is safe for handling and applicable for agricultural use.

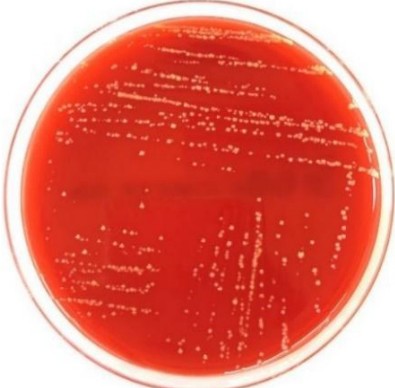

**Figure 2.** Hemolysis test carried out on *S. yanoikuyae* SJTF8 showed no hemolysis by bacteria.

### 3.3. Phosphate Solubilization

Phosphorus (P) is the second most important element after N, and it plays an integral role in the biological processes and physiological development of plants. Phosphorus is a key component in Adenosine 5′-triphosphate (ATP), glucose, and nucleic acids, and it also facilitates the phosphodiester bonds that bind amino acids together. P is mostly absorbed in the form of phosphates ($HPO_4^{2-}$ and $H_2PO_4^{-}$). The phosphates that are present in the soil are unavailable to plants. The natural P fixation processes differ in soils and are closely related to the soil pH. One biological process of P fixation is through the actions of soil microorganisms. Besides fungus, some bacteria species participate in mineralization processes, whereby insoluble soil nutrients are transformed into available forms. These bacteria species are recognized for their role and are referred to as PGPR or P-solubilization bacteria (PSB).

We lack adequate information on the ability of *S. yanoikuyae* to solubilize P. In this study, *S. yanoikuyae* SJTF8 demonstrated its ability to solubilize P, which we qualitatively characterized by the clear zones (halo zones) around the microbial colonies growing on the Pikovskaya agar (Figure 3). Adequate amounts of P are vital for photosynthesis, flowering, fruiting, seed production, and plant maturation. The bacteria also influence the root growth, and especially the development of the lateral and fibrous rootlets. A deficiency of the nutrient may lead to stunted plant growth, which is characterized by thin and spindly stumps, poor root growth, and dark-green foliage. As for rice plants, P deficiency may also delay the crop maturation and produce high levels of unfilled grain.

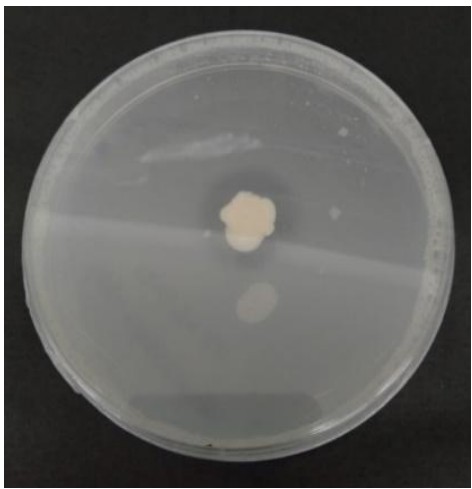

**Figure 3.** Halo zone on Pikovskaya agar indicating *S. yanoikuyae* SJTF8 phosphate solubilization.

### 3.4. Effects of S. yanoikuyae SJTF8 on Seed Germination and Root and Shoot Lengths

The seedling development stage is a vital phase in the lives of plants. Taiwanese rice farmers tend to transplant rice seedlings from 15 to 21 days after germination. Thus, observing the bacterial effect on the seedling growth and the amount of IAA present in the seedling prior to transplant can provide options to ensure better crop growth. This early growth stage can also encourage the plant biomass production, which can substantially influence the grain production [37]. The application of *S. yanoikuyae* SJTF8 with different concentrations had varied effects on the rice seedling growth. Compared with the untreated control, inoculating rice seeds with *S. yanoikuyae* SJTF8 dramatically enhanced the rice seed germination rate and vigor index (Table 2).

**Table 2.** Mean comparison of effects of different concentrations of *S. yanoikuyae* SJTF8 on rice seed germination percentage, speed of germination, and germination vigor.

| Treatment | Germination | Speed of Germination | Seed Germination Vigor |
|---|---|---|---|
| | % | No. Seeds/Day | |
| Control | 82.50 b $\pm$ 9.57 | 5.78 b $\pm$ 1.08 | 301.46 b $\pm$ 28.69 |
| SP-y 4 | 90.00 b $\pm$ 8.16 | 6.10 ab $\pm$ 0.88 | 329.23 b $\pm$ 29.24 |
| SP-y 5 | 90.00 ab $\pm$ 8.16 | 6.25 ab $\pm$ 0.64 | 399.95 ab $\pm$ 30.36 |
| SP-y 6 | 100.00 a $\pm$ 0.00 | 6.78 ab $\pm$ 0.76 | 456.65 a $\pm$ 34.78 |
| SP-y 7 | 100.00 a $\pm$ 0.00 | 7.08 a $\pm$ 0.29 | 453.05 a $\pm$ 14.17 |
| SP-y 8 | 92.50 ab $\pm$ 15.00 | 6.48 ab $\pm$ 0.63 | 347.68 ab $\pm$ 58.23 |
| Av. | 92.5 | 6.41 | 381.34 |

The abbreviations given to the isolates are as follows: control: without bacteria; SP-y 4: $10^4$ CFU/mL; SP-y 5: $10^5$ CFU/mL; SP-y 6: $10^6$ CFU/mL; SP-y 7: $10^7$ CFU/mL; SP-y 8: $10^8$ CFU/mL. Mean values followed by different letters within columns significantly differ at $p \leq 0.05$ according to DMRT. We present the standard deviation after the $\pm$symbol.

The germination rates were high in the rice seeds treated with *S. yanoikuyae* SJTF8 at the SP-y 7 and SP-y 6 concentrations, with the best germination rate at 100%, compared with the untreated control seeds, which had 80% germination. We observed a progressive increase in the vigor index, which reflects the plant's ability to compete for light, nutrients, air, and water, as well as to adequately survive under stress [49]. We obtained significant treatment ($p \leq 0.05$) effects in the SP-y 7 and SP-y 6 concentrations (453.05 and 456.65, respectively), which were related to the seed germination rates and significant root lengths ($p \leq 0.05$) of the two respective treatments (9.23 cm and 8.55 cm, respectively) (Tables 2 and 3, respectively). The germination speeds in most of the treatments were not substantially different. We only observed a significant difference ($p \leq 0.05$) in the SP-y 7 treatment (7.08 seeds/day) and control treatment (5.78 seeds/day).

**Table 3.** Mean comparison of effects of six concentration levels of *S. yanoikuyae* SJTF8 on rice root lengths from 6 to 14 days after germination.

| Treatment | 6th Day after Germination | 9th Day after Germination | 12th Day after Germination | 14th Day after Germination |
|---|---|---|---|---|
| | cm | | | |
| Control | 3.55 a $\pm$ 2.11 | 4.38 c $\pm$ 2.59 | 5.31 c $\pm$ 3.14 | 5.99 c $\pm$ 3.40 |
| SP-y 4 | 3.64 a $\pm$ 2.46 | 5.38 abc $\pm$ 3.35 | 6.54 bc $\pm$ 3.71 | 7.62 b $\pm$ 4.03 |
| SP-y 5 | 4.00 a $\pm$ 1.84 | 6.32 a $\pm$ 2.82 | 7.11 ab $\pm$ 3.00 | 8.15 ab $\pm$ 3.31 |
| SP-y 6 | 4.43 a $\pm$ 2.07 | 6.09 ab $\pm$ 2.42 | 7.50 ab $\pm$ 2.42 | 8.56 ab $\pm$ 2.31 |
| SP-y 7 | 4.53 a $\pm$ 1.39 | 6.56 a $\pm$ 2.02 | 8.32 a $\pm$ 2.30 | 9.29 a $\pm$ 2.25 |
| SP-y 8 | 3.66 a $\pm$ 2.00 | 4.89 bc $\pm$ 2.33 | 6.57 bc $\pm$ 2.90 | 7.68 b $\pm$ 3.29 |
| Av. | 3.97 | 5.60 | 6.89 | 7.88 |

Mean values followed by different letters within columns significantly differ at $p \leq 0.05$ according to DMRT. We present the standard deviations after the $\pm$symbol.

The first stage of plant growth and development is seed germination, and it is also the most vulnerable stage in the plant lifecycle. In this stage, the seed embryo's key components develop, and it demonstrates its potential to germinate, which is marked by the appearance of radicals that penetrate the seed coat. Well-maintained seedlings eventually grow into healthy plants that produce biomass that can influence the crop's productivity. The significant ($p \leq 0.05$) improvement in the seed germination, root length, and vigor index might be due to the production of IAA by the presence of rhizobacteria. These findings are congruent with those reported by Spaepen and Vanderleyden [50], who proved that, among the auxins, IAA is the most frequent physiologically active plant hormone, regulating different aspects of plant growth and development. The longest root

length was produced by Sp-y 7, and the lowest was produced by the control treatment, between the 9th and 14th days after germination (Table 3).

We present the effects of the different concentrations of *S. yanoikuyae* SJTF8 on the shoot lengths of the rice seedlings in Table 4. From the 9th to 12th days, the shoot lengths of the inoculated seedlings significantly ($p \leq 0.05$) increased compared with the controlled seedlings. After the 12th and 14th days, the seedlings inoculated with SP-y 6 produced the highest shoot length values among the other treatments (Table 4). In general, we determined that the SP-y 6 and SP-y 7 treatments promoted better shoot development after germination. Furthermore, the seedling rootlets had substantial increases in their lengths on the 9th and 14th days after germination. In general, the application of the bacteria on the root and shoot lengths showed that *S. yanoikuyae* SJTF8 has the potential to stimulate growth in rice seedlings. In a previous study in Taiwan, researchers isolated a subspecies of *S. yanoikuyae* along with two other bacteria species. When combined, they were able to promote growth in rice plants under depleted soil moisture conditions. The combination of PGPR also allowed for a reduction in the synthetic fertilizer application of up to 50% [37,38].

**Table 4.** Mean comparison of effects of six concentration levels of *S. yanoikuyae* SJTF8 on rice shoot lengths from 6 to 14 days after germination.

| Treatment | 6th Day after Germination | 9th Day after Germination | 12th Day after Germination | 14th Day after Germination |
|---|---|---|---|---|
| | | cm | | |
| Control | 1.44 a ± 0.92 | 3.15 b ± 1.78 | 4.40 c ± 2.62 | 5.60 b ± 3.50 |
| SP-y 4 | 1.54 a ± 1.02 | 3.61 ab ± 2.16 | 5.13 bc ± 2.78 | 7.27 a ± 3.92 |
| SP-y 5 | 1.75 a ± 1.00 | 4.24 a ± 1.83 | 5.46 abc ± 2.44 | 7.21 a ± 3.80 |
| SP-y 6 | 1.89 a ± 1.03 | 4.27 a ± 1.37 | 6.36 a ± 2.04 | 7.68 a ± 2.25 |
| SP-y 7 | 1.90 a ± 0.70 | 3.97 ab ± 1.44 | 5.82 ab ± 1.18 | 7.55 a ± 1.90 |
| SP-y 8 | 1.63 a ± 0.96 | 3.47 ab ± 1.81 | 5.29 abc ± 2.42 | 6.68 ab ± 3.05 |
| Av. | 1.69 | 3.79 | 5.41 | 7.00 |

Mean values followed by different letters within columns significantly differ at $p \leq 0.05$ according to DMRT. We present the standard deviation after the ±symbol.

### 3.5. Physical Observations of Effects of S. yanoikuyae SJTF8 on Rice Root Systems

The root hair development was promoted by the bacteria (Figure 4). The seedlings under the control treatment had minimal root hair growth (Figure 4a). The seedlings inoculated with the bacteria *S. yanoikuyae* SJTF8 had dense root hair development, irrespective of the treatment. We observed that the seedlings treated with the SP-y 6 and SP-y 7 concentrations produced the highest root hair densities (Figures 4d and 4e, respectively). We observed an accumulation of bacteria at the basal section of the root hair, and the creation of microaggregates on the epidermal surface (Figure 5). Microorganisms, such as bacteria, invade and colonize damaged epidermal tissues [14,15,51]. Perhaps the buildup is induced by the actions of microbial activities or by mechanical damage. The mutualistic interactions between plants and microorganisms can improve the nutrient availability and absorption for plant growth. According to Figures 5a and 6a, there was no bacterial colonization in the uninoculated seedlings. The seedlings inoculated with *S. yanoikuyae* SJTF8 had bacterial colonization on the root surface (Figure 6). Microbes not only accumulate on the plant's outer surfaces as epiphytes, but also within plant tissues as endophytes [52]. The presence of these microbial communities both on and within a plant root may influence the host plant's physiological function and development [53]. Endophytes create phytohormones, or alter their levels, which suggests that they may play a role in plant growth [54]. Moreover, these bacteria species are known to transport nutrients to the plant roots [55,56].

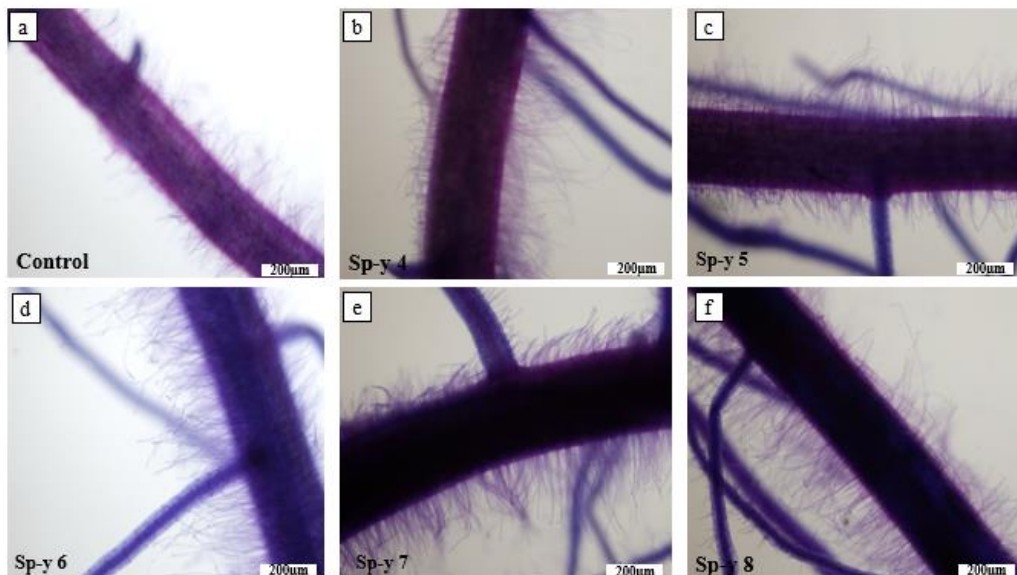

**Figure 4.** Effects of *S. yanoikuyae* SJTF8 at different CFU/mL concentrations on development of root hairs of 10-day-old rice seedlings. We stained the rootlets of the seedlings to make the root hairs visible. (**a**) Root hairs of 10-day-old seedlings under control. (**b**) Treatment effect showing visible increase in the root hair of seedlings treated with SP-y 4; (**c**) Treatment effect showing an increase in the density of root hair seedlings treated with SP-y 5; (**d**) Treatment effect showing a notable increase in the root hair length of seedlings treated with SP-y 6; (**e**) Visual observation of the root hair of rice seedlings treated with SP-y 7; (**f**) The root hair of rice seedlings treated with SP-y 8.

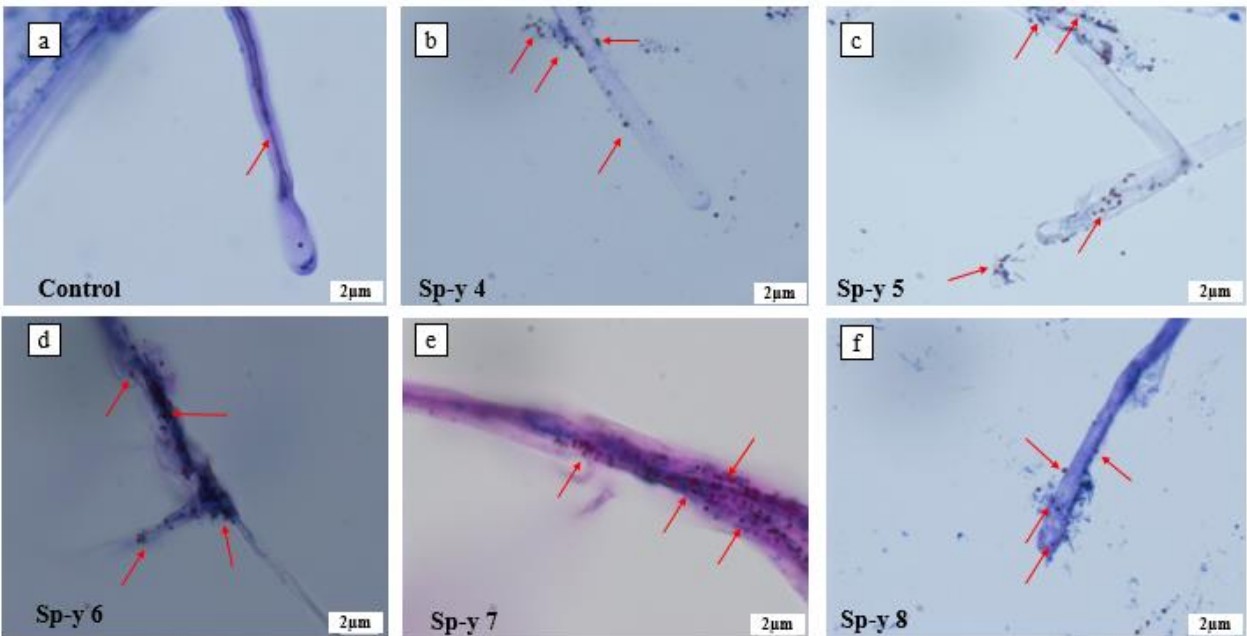

**Figure 5.** Microscopic visualization of rice hair surface with the presence of *S. yanoikuyae* SJTF8. (**a**) Root hairs of 10-day-old seedlings under control absent of bacteria. (**b**) Root hairs of 10-day-old seedlings treated with SP-y 4 showing the presence of the bacteria; (**c**) An increase in bacterial presence in the root hair of seedlings treated with SP-y 5; (**d**) A wide distributing of bacteria on the root hair of rice seedlings treated with SP-y 6; (**e**) Rice root hair treated with SP-y 7 also with a wide distribution of bacteria; (**f**) Notable presence of the bacteria on the root hair surface of rice seedlings treated with SP-y 8.

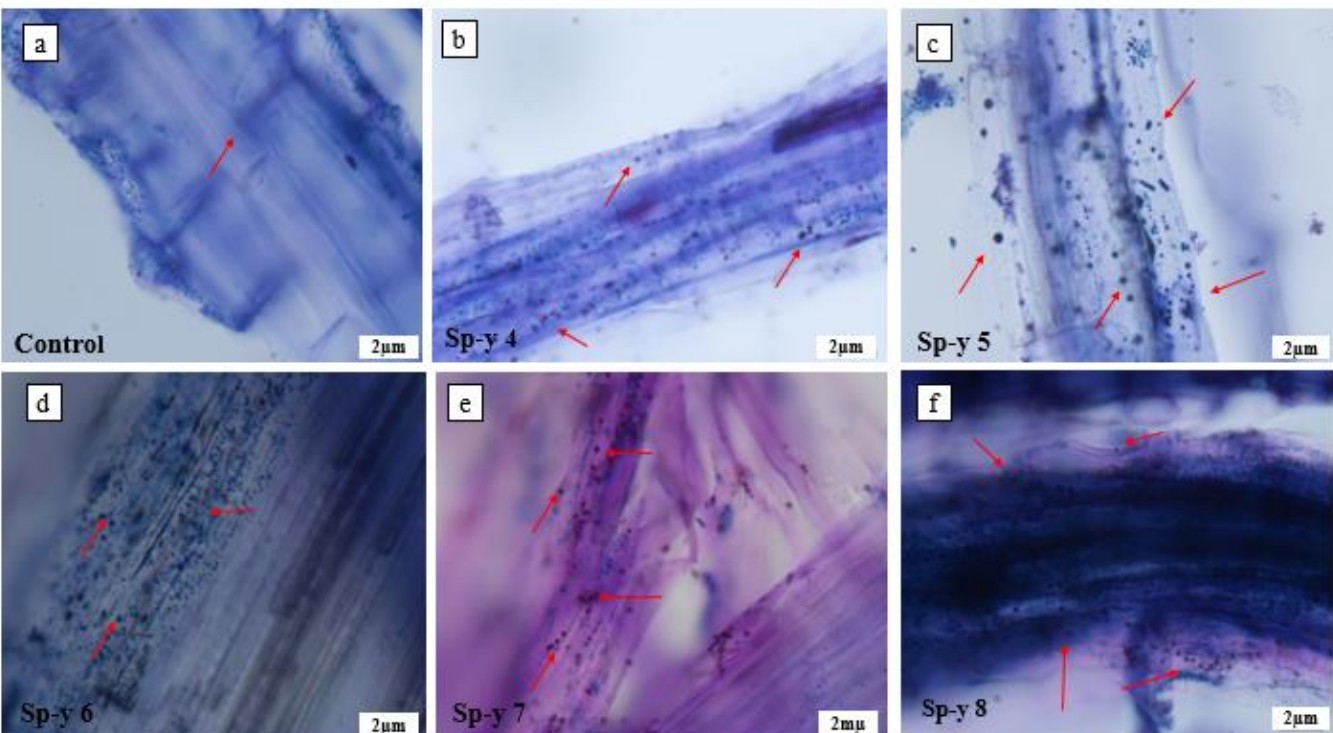

**Figure 6.** Microscopic images of root tissues of rice seedlings revealing presence of bacteria cells. (**a**) The root tissue of 10-day-old seedlings under the control treatment. (**b**) Visible presence of bacteria present in the root tissue of rice seedlings treated with SP-y 4; (**c**) Bacteria present in the root tissue of seedlings treated with SP-y 5; (**d**) Rice seedlings treated with SP-y 6 with high presence of the bacteria in the root tissue; (**e**) Root tissue of rice seedlings treated with SP-y 7; (**f**) Bacteria present in seedlings treated with SP-y 8.

### 3.6. Indole Acetic Acid Production

The plant growth cycle is determined by the chemical compounds that are categorized as phytohormones. IAA is present in the form of auxins and is a vital molecule that regulates specialized plant growth. It facilitates plant cell elongation by modifying certain conditions, such as increasing the cell's osmotic content; increasing the water absorption by the cell; decreasing the wall pressure; increasing the cell wall synthesis; inducing protein synthesis; inhibiting or delaying leaf abscission; inducing flowering and fruiting [57]. The auxins, along with other plant phytohormones, are inactive when the seeds are in the dormant stage. Seed germination is predominately triggered by the hormone gibberellin (GA). Plant phytohormones, such as auxins and cytokinins, become active once the seed dormancy is terminated. The phytohormone ethylene is also induced, but at a later growth stage. These phytohormones naturally occur in low concentrations in plants. Thus, the presence of PGPR may have the potential to influence the occurrence of these chemical compounds at an early stage of the plant's life.

The bacteria *S. yanoikuyae* SJTF8 may have some influence over the hormone GA, which we speculated from the speed of the germination (Table 2). Lee et al. [58] reported a similar finding. They found that the tested PGPR strain was able to increase the root and shoot lengths, along with the dry weights, of 16-day-old seedlings. According to Pirilak et al. [59] and Esitken et al. [60], GAs have a similar effect as auxins and cytokinins on plant growth. They are produced in the meristematic tissues of the shoots and roots, and they increase the shoot elongation. Taiz and Zeiger [61] and Stowe and Yamaki [62] have further suggested that GAs participate in internode elongation, pollen tube growth, and floret development.

We processed the IAA concentrations using an ELISA kit. The kit is a competitive inhibition enzyme immunoassay technique for the in vitro quantitative measurement of

IAA in tissue. Some rhizobacteria species can produce auxins, and they thus affect the auxin levels in plant roots [63,64]. *S. yanoikuyae* is less known for its capacity to stimulate the synthesis of IAA. Chen et al. [29] reported that the existence of *S. yanoikuyae* Sy310 stimulated higher IAA production, in addition to other biosynthesis activities, in the host plants. These increases in the IAA production increased the root and plant biomass. Similarly, in a study on Chinese cabbage (*Brassica campestris* ssp. *Pekinensis*) inoculated with *S. yanoikuyae*, the researchers found that the bacteria were able to trigger higher IAA production compared with some of the other bacteria species used in the study, which suggests that each treatment influenced the amount of IAA that occurred in the seedling root tissue (Figure 7). The rice seedlings treated with the SP-y 7 and SP-y 8 concentrations produced the highest IAA levels among all the treatments, ranging from 151,029 pg/mL to 168,033 pg/mL. The amount of IAA produced is important because it can also modify the plant's physical appearance, such as by enhancing the length and density of the seedling root system (Table 3 and Figure 4) [57]. In turn, this aids the host plant in maximizing its nutrient and water uptake.

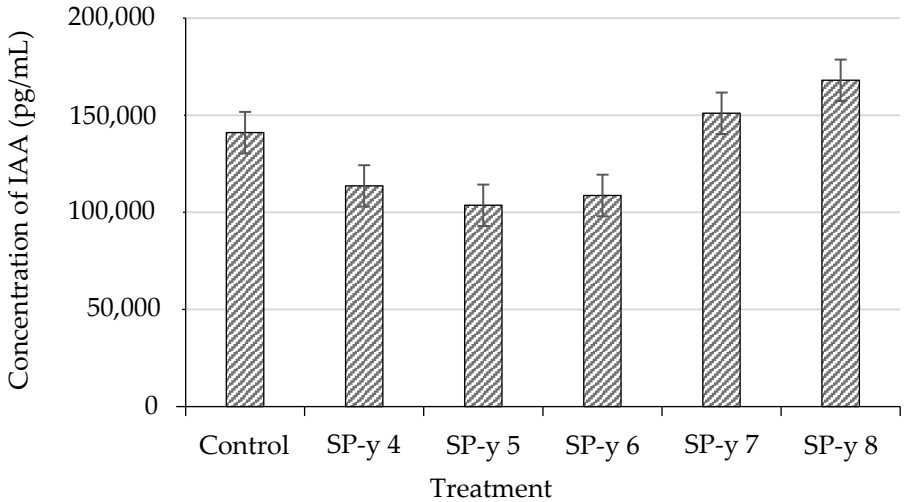

**Figure 7.** Indole-3-acetic acid (IAA) concentrations produced by rice seedlings inoculated with *S. yanoikuyae* SJTF8 at different CFU/mL levels.

The different CFU levels from the treatments applied may have some degree of influence on the amount of IAA that is present in the seedling tissue (Figure 7). For instance, the seedlings treated with the SP-y 7 and SP-y 6 concentrations produced significantly better ($p \leq 0.05$) seedling growth compared with those treated with the SP-y 8 concentration. According to Spaepen and Vanderleyden [50], the amount of IAA present in the plant tissue can modify its root growth, cell division, and development. Based on this notion, it is assumed that a proper balance of IAA is required to improve the seedling growth prior to transplanting. Thus, in this study, we propose that SP-y 7 and SP-y 6 are the most suitable CFU/mL concentrations for seed inoculation.

## 4. Conclusions

Inoculating rice seedlings with *S. yanoikuyae* SJTF8 considerably enhanced the germination rate, germination percentage, seed vigor index, and seedling root and biomass. The bacteria demonstrated some characteristics that could be beneficial for agricultural use. As observed from the germination processes and seedling growth and development, the bacteria had the ability to solubilize P and synthesize IAA. Higher levels of inoculates produced higher amounts of IAA. From the treatments tested, the SP-y 7 and SP-y 8 concentrations produced the highest amounts of IAA. However, we consider the conducive concentrations for *S. yanoikuyae* SJTF8 to be SP-y 7 and SP-y 6. The association of *S. yanoikuyae* SJTF8 with rice was found to trigger the growth and development of the crop root system at the early

stage of growth. Much is yet to be understood about the bacterium; however, according to the present results, it has the potential to be utilized as a PGPR source for sustainable rice production.

**Author Contributions:** Conceptualization, Y.-T.J. and Y.-M.W.; methodology, Y.-T.J. and Y.-M.W.; software, E.J.T. and C.K.K.; validation, Y.-T.J. and Y.-M.W.; formal analysis, Y.-T.J., E.J.T. and C.K.K.; investigation, E.J.T., C.K.K. and Y.-T.W.; resources, Y.-T.J. and Y.-M.W.; data curation Y.-T.J. and E.J.T.; writing—original draft preparation, E.J.T. and C.P.; writing—review and editing, C.K.K., Y.-T.J., Y.-T.W. and Y.-M.W.; visualization, Y.-T.J. and Y.-M.W.; supervision, Y.-T.J., C.P. and Y.-M.W. All authors have read and agreed to the published version of the manuscript.

**Funding:** COA grant no. 109AS-7.1.3-IE-b2.

**Institutional Review Board Statement:** Not applicable.

**Data Availability Statement:** The data presented in this article are available in the NPUST library database.

**Acknowledgments:** The authors want to acknowledge the collaboration between the students and staffers of Brawijaya University, Indonesia, and of the National Pingtung University of Science and Technology, Taiwan (R.O.C).

**Conflicts of Interest:** The authors declare no conflict of interest.

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
