# Peer review of "Effects of Sphingobium yanoikuyae SJTF8 on Rice (Oryza sativa) Seed Germination and Root Development"

_agriculture, doi:10.3390/agriculture12111890_

Round 1

Reviewer 1 Report (Previous Reviewer 2)

A lot of improvement compared to the previous submission.

In Result and Discussion section = How do you compare your results (especially in root and shoot length at the seedling stage under S. yanoikuyae treatments) with the chemical fertilizers.

Author Response

Greetings,

Thank you for your comments and suggestions. Please find below our response to your queries. The revised draft will be submitted in PDF format.

Comments and Suggestions for Authors

Response

In Result and Discussion section = How do you compare your results (especially in root and shoot length at the seedling stage under S. yanoikuyae treatments) with the chemical fertilizers.

Please refer to Lines 333-337 of the revised manuscript.

Reviewer 2 Report (New Reviewer)

This manuscript reported on the effect of a bacterium, Sphingobium yanoikuyae on seed germination and root development of rice. The content of the manuscript fits well with the aim and scope of the journal. Overall quality of the manuscript is average. The quality of English usage must be improved. Therefore, I recommend a major revision of this manuscript in its current format. Some concerns are listed below. 

Abstract 

Line 22 specify the bacteria 

Line 25 delete Sp-y 

Materials and methods 

Line 114 high yield aromatic rice 

Isolation of bacteria: how many bacteria were isolated? What is the isolation medium? Why reported only one isolate, SJTF8? 

Line 124-125 “…and single colony isolate at -80 oC in the freezer.” This is not understandable. 

Identification of bacteria: Please use EzBioCloud website for BLAST analysis 

Line 145 solubilize phosphate to phosphorus 

Line 153 The seeds…. 

Seed germination: how many days for germination experiment? when germination was recorded? 

Line 198-199    Please clarify “The wavelength was later expended to 570 nm or 630 nm to account for any errors.” Rationale of 570/630 nm can identify error of 450 nm 

Results and Discussion 

There is no result for bacterial isolation 

Reanalysis BLAST result using EzBioCloud database 

Line 230 Gram negative 

Line 246-247 …did not break red blood cells 

For safety issue of the proposed strain, the authors are strongly recommended to provide evidence on the safety of the proposed strains and add discussion regarding their safety. Please consider these references as a guideline. 

1. Keswani, C., Prakash, O., Bharti, N., Vilchez, J.I., Sansinenea, e., Lally, R.D., Borriss, R., Singh, S.P., Gupta, V.K., Fraceto, L.F., de Lima, R. and Singh, H.B. 2019. Re-addressing the biosafety issues of plant growth promoting rhizobacteria. Science of the Total Environment 690: 841-852. 

2. Barros-Rodriguez, A., Rangseekaew, P., Lasudee, K., Pathom-aree, W. and Manzanera, M.  2020. Regulatory risks associated with bacteria as biostimulants and biofertilizers in the frame of the European regulation (EU) 2019/1009. Science of the Total Environment 740: 140239. 

3. Rangseekaew, P., Barros-Rodriguez, A., Pathom-aree, W. and Manzanera, M. 2021. Deep-sea actinobacteria mitigate salinity stress in tomato seedlings and their biosafety testing. Plants 10(8): 1687. https://doi.org/10.3390/plants10081687 

4. Rangseekaew, P., Barros-Rodriguez, A., Pathom-aree, W. and Manzanera, M. 2022. Plant beneficial deep-sea actinobacterium, Dermacoccus abyssi MT1.1T promote growth of tomato (Solanum lycopersicum) under salinity stress. Biology 11(2): 191. https://doi.org/10.3390/biology11020191 

Line 256 ability to dissolve PO4 to P 

Table 2 there is no unit for speed of germination? 

Why concentration at 108 cfu/ml did not insert any positive effect on rice? Please explain this point. 

Line 172 highest… 

Line 297 4.35 cm but in table 4 was 4.24 

Line 323-324 “Such epidermal surface damage in strongly colonized locations shows an aggressive invasion process” with such aggressive invasion, how rice recognized strain SJTF8 as beneficial microorganism? Please discuss this point. 

Line 341 why fix N? 

Line 354 why reported IAA concentration as pg? 

Why in vitro determination of IAA production was not carried out in SJTF8? 

Since plant is also able to produce IAA, how can the authors differentiate between IAA produced by plant and strain SJTF8? 

The experiment is carried out for 14 days considering the whole rice cultivation time is generally 90-120 days. Is this period of time suitable to provide representative answer to the reseach question being asked? Please provide evidence support the rationale of 14 days experiment. 

Line 400 “The bacterium is found to be safe for agricultural use” This sentence was over statement and speculative as only hemolytic test was done.            

Author Response

Greetings,

Thank you for your comments and suggestions. Please find below our response to your queries. Please find attached the revised manuscript in PDF format.

Comments and Suggestions for Authors

Response/action

Line 22 specify the bacteria 

Edited accordingly (Line 21)

Line 25 delete Sp-y 

The section has been revised (Lines 26-27)

Line 114 high yield aromatic rice 

Edited accordingly (Line 107)

Isolation of bacteria: how many bacteria were isolated? What is the isolation medium? Why reported only one isolate, SJTF8? 

A description was provided from lines 115-116 and in Table 1 (Lines 231 to 238).

Line 124-125 “…and single colony isolate at -80 oC in the freezer.” This is not understandable.

The section has been revised (Lines 119-120)

Identification of bacteria: Please use EzBioCloud website for BLAST analysis 

The identification process was carried out by Food Industry Research and Development Institute, Hsinchu, Taiwan (ROC). Thus, the result provided are in line with the NCBI database (Line 124-125)

Line 145 solubilize phosphate to phosphorus 

Revised (Line 146)

Line 153 The seeds…. 

Edited accordingly (Line 147)

Seed germination: how many days for germination experiment? when germination was recorded? 

The section has been revised (Lines 212-216)

There is no result for bacterial isolation 

The section has been revised with additional information provided (Lines 232-238)

Reanalysis BLAST result using EzBioCloud database 

The identification process was carried out by Food Industry Research and Development Institute, Hsinchu, Taiwan (ROC). Thus, the result provided was in line with the NCBI database (Line 124-125). 

Line 230 Gram negative 

Edited accordingly (Line 241)

Line 246-247 …did not break red blood cells 

Edited accordingly (Line 260)

1. Keswani, C., Prakash, O., Bharti, N., Vilchez, J.I., Sansinenea, e., Lally, R.D., Borriss, R., Singh, S.P., Gupta, V.K., Fraceto, L.F., de Lima, R. and Singh, H.B. 2019. Re-addressing the biosafety issues of plant growth promoting rhizobacteria. Science of the Total Environment 690: 841-852. 

Cited (Line 253-255)

2. Barros-Rodriguez, A., Rangseekaew, P., Lasudee, K., Pathom-aree, W. and Manzanera, M.  2020. Regulatory risks associated with bacteria as biostimulants and biofertilizers in the frame of the European regulation (EU) 2019/1009. Science of the Total Environment 740: 140239. 

Cited (Line 253-255)

Line 256 ability to dissolve PO4 to P 

The study would like to observe if the strain was able to utilize P from the agar thus indicating its ability to solubilize P to PO4

Table 2 there is no unit for speed of germination? 

In No. of seeds germinated/day. Refer to Lines 298-299)

Why concentration at 108 cfu/ml did not insert any positive effect on rice? Please explain this point. 

An explanation was provided (Lines 399-405).

Line 172 highest…

Edited accordingly

Line 297 4.35 cm but in table 4 was 4.24 

The section has been revised (Lines 325-333)

Line 323-324 “Such epidermal surface damage in strongly colonized locations shows an aggressive invasion process” with such aggressive invasion, how rice recognized strain SJTF8 as beneficial microorganism? Please discuss this point. 

The section has been revised (Lines 343-354)

Line 341 why fix N? 

The section has been revised (Line 374-375).

Line 354 why reported IAA concentration as pg? 

The unit of measurement was revised to pg/mL while one of the graphs have been omitted

Why in vitro determination of IAA production was not carried out in SJTF8? 

The section has been revised with additional information provided (Lines 372 to 374)

Since plant is also able to produce IAA, how can the authors differentiate between IAA produced by plant and strain SJTF8? 

The section on IAA production was revised to accommodate for the suggestions/questions (Lines 358-406)

The experiment is carried out for 14 days considering the whole rice cultivation time is generally 90-120 days. Is this period of time suitable to provide representative answer to the reseach question being asked? Please provide evidence support the rationale of 14 days experiment

The section has been revised with additional information provided (Lines 290-296)

Line 400 “The bacterium is found to be safe for agricultural use” This sentence was over statement and speculative as only hemolytic test was done

The section has been revised (Lines 412-413)

Best regards

Reviewer 3 Report (New Reviewer)

Review on manuscript agriculture-1937050

Effects of Sphingobium yanoikuyae SJTF8 on Rice (Oryza sativa) Seed Germination and Root Development

The paper is novel in some respects and has some interesting and complete results that are of interest. However, some questions of methodology and order and prioritization in the text must be improved. Also, the main concern is in the English text, it should be improved and ordered before being considered for publication. Some methodological aspects need to be improved. A list of suggestions to improve the manuscript is provided.

Some comments and suggestions:

50 - … important, [7]. … delete comma.

57 - … pathogenic induces., pathogenic inducers?

58 - … plant growth-promorting rhizobacteria (PGPR), correct plant growth-promoting rhizobacteria.

62 - … are of the genera, … or are in the genera?

65 - … through either direct or indirect mechinzation, please rephrase.

71 - Capitalizing on these effects can contribute emmursly towards, … please rephrase.

73 – 74 – “A less known bacteria species in the field of agriculture is the Sphingobium yanoikuyae SJTF8 from the genera Sphingomonas under the class Alphaproteobacteria”. This is a confusing sentence, If the name of the species of bacteria is Sphingobium yanoikuyae, the genus is Sphingobium, not Sphingomonas, but was originally described in Sphingomonas (Yabuuchi et al. 1990). Also, rephrase.

76 - Early studies into the genera… what genera? In the previous sentence was named a genus.

77 - Various Sphingomonas sp … sp. denotes singular, various species must be written in plural spp.

81 – … potrays?, … portrays?

90 - … moisture deficite condition … deficit …, I will not comment further on the many small grammatical or punctuation errors.

92 – 94 - The result showed that the combination triggeed … the sentence contains several errors, it must be rewritten.

121 – What is LB? Liquid broth? Do not leave known terminology of the discipline as obvious.

124 – 125 – “Subculture once a month for the duration of the study, and single colony isolate at -80 oC in the freezer” … please rephrase.

126 - 2.3. Identification Bacteria … Strain identification.

140 - … a single colonies … a single colony or single colonies?

149 - … by identifying the halo zone around the S. yanoikuyae bacterial colonies …only identified or also measured?

151 - 2.6. Sterilization of Seeds … Seed Surface sterilization.

168 – Each replicate per treatment consisted of 10 rice seeds of KH-147. Why was a germination test done with only 10 seeds? Generally 50-100 seeds are used in this type of test, and there are even protocols with 400 seeds, which does not seem difficult to carry out in germination tests.

182 - … The mixture was centrifuge at 2.000 g, …centrifuged, also review units and abbreviations.

199 -200 - … A standard curve was generated to measure the amount of IAA present in the 199 plants tissue (Figure 1). Remove to Results section with the Figure.

211 – … Formula (3) … seed vigor.

216 - … Baso Rapid Gram Stain … provide company or composition?

219 - … The IBM SPSS was used to perform … program?

221-222- …Why was a Duncan test used? Although not incorrect, for most laboratory analysis the Tuckey test is used.

225 - … The bacterium identification was carried out by DNAeasy Plant Kit (Mission Biotech, Taiwan) and the gentic sequence reference was confirmed with NCBI … very confusing sentence. The bacterial identification was not carried out with the DNAeasy kit, the kit was used for DNA extraction and the “genetic” sequence compared with sequences in NCBI database. Please, rephrase the sentence. As understand, the strain was sequenced and compared with reference sequences in NCBI.

228 – The sequence of the strain used in this paper has a 100% identical identity with the 16s gene of the type strain of this species? Is that comparison enough to say that it is the type strain SJTF8?

228 - … From the Sphingomonas genus in is found unde the class Alphaproteobacteria which are widely known to be strictly aerobic … very poorly written sentence, please rewrite.

230 - … gram-negetive … gram negative.

224 – Rephase all the 3.1 Identification of bacteria paragraph.

245 - … The measurement of the existence of hemolysis has can indicate suspected … rephrase.

263 - Only a small halo is seen as an indication of phosphorus solubilization. it is not possible to adjust this method to make it semi-quantitative and more accurate? For example, by measuring halo diameter in several repetitions.

270 – Table 2 … the percentage of seed germination was calculated after 10 seeds were planted?

350 - … The result showed that IAA was synthesis … synthesized …

And no more comments for the moment.

Author Response

Greetings,

Thank you for your comments and suggestions. Please find below our response to your queries. Please find attached the revised manuscript in PDF format.

Comments and Suggestions for Authors

Response/action

50 - … important, [7]. … delete comma.

Edited accordingly

57 - … pathogenic induces., pathogenic inducers?

Revised

58 - … plant growth-promorting rhizobacteria (PGPR), correct plant growth-promoting rhizobacteria.

Edited accordingly

62 - … are of the genera, … or are in the genera?

The section has been revised

65 - … through either direct or indirect mechinzation, please rephrase

The section has been revised

71 - Capitalizing on these effects can contribute emmursly towards, … please rephrase

The section has been revised

73 – 74 – “A less known bacteria species in the field of agriculture is the Sphingobium yanoikuyae SJTF8 from the genera Sphingomonas under the class Alphaproteobacteria”. This is a confusing sentence, If the name of the species of bacteria is Sphingobium yanoikuyae, the genus is Sphingobium, not Sphingomonas, but was originally described in Sphingomonas (Yabuuchi et al. 1990). Also, rephrase.

The section has been revised

76 - Early studies into the genera… what genera? In the previous sentence was named a genus.

The section has been revised

77 - Various Sphingomonas sp … sp. denotes singular, various species must be written in plural spp.

The section has been revised

81 – … potrays?, … portrays?

Edited accordingly

90 - … moisture deficite condition … deficit …, I will not comment further on the many small grammatical or punctuation errors

Edited accordingly

92 – 94 - The result showed that the combination triggeed … the sentence contains several errors, it must be rewritten

The section has been revised

121 – What is LB? Liquid broth? Do not leave known terminology of the discipline as obvious.

The section has been revised

124 – 125 – “Subculture once a month for the duration of the study, and single colony isolate at -80 oC in the freezer” … please rephrase

The section has been revised

126 - 2.3. Identification Bacteria … Strain identification.

The section has been revised

140 - … a single colonies … a single colony or single colonies?

The section has been revised

149 - … by identifying the halo zone around the S. yanoikuyae bacterial colonies …only identified or also measured?

Only qualitative analysis was carried out to observe if the bacteria has the potential to solubilize P.

151 - 2.6. Sterilization of Seeds … Seed Surface sterilization

Edited accordingly

168 – Each replicate per treatment consisted of 10 rice seeds of KH-147. Why was a germination test done with only 10 seeds? Generally 50-100 seeds are used in this type of test, and there are even protocols with 400 seeds, which does not seem difficult to carry out in germination tests.

Each treatment contains 4 replicates. Thus, sample number is 40.

182 - … The mixture was centrifuge at 2.000 g, …centrifuged, also review units and abbreviations.

The section has been revised

199 -200 - … A standard curve was generated to measure the amount of IAA present in the 199 plants tissue (Figure 1). Remove to Results section with the Figure.

The graph was omitted from the paper

211 – … Formula (3) … seed vigor.

Edited accordingly

216 - … Baso Rapid Gram Stain … provide company or composition?

The section has been revised with additional information provided

219 - … The IBM SPSS was used to perform … program?

The section has been revised with additional information provided

221-222- …Why was a Duncan test used? Although not incorrect, for most laboratory analysis the Tuckey test is used.

Because it is widely used in other published studies. Also, sed DMRT because it is slightly sensitive then Turkey test.

225 - … The bacterium identification was carried out by DNAeasy Plant Kit (Mission Biotech, Taiwan) and the gentic sequence reference was confirmed with NCBI … very confusing sentence. The bacterial identification was not carried out with the DNAeasy kit, the kit was used for DNA extraction and the “genetic” sequence compared with sequences in NCBI database. Please, rephrase the sentence. As understand, the strain was sequenced and compared with reference sequences in NCBI.

The section has been revised

228 – The sequence of the strain used in this paper has a 100% identical identity with the 16s gene of the type strain of this species? Is that comparison enough to say that it is the type strain SJTF8?

A description was provided from lines 115-116 and in Table 1 (Lines 231 to 238).

228 - … From the Sphingomonas genus in is found unde the class Alphaproteobacteria which are widely known to be strictly aerobic … very poorly written sentence, please rewrite.

The section has been revised

230 - … gram-negetive … gram negative

Edited accordingly

224 – Rephase all the 3.1 Identification of bacteria paragraph.

Edited accordingly

245 - … The measurement of the existence of hemolysis has can indicate suspected … rephrase.

The section has been revised

263 - Only a small halo is seen as an indication of phosphorus solubilization. it is not possible to adjust this method to make it semi-quantitative and more accurate? For example, by measuring halo diameter in several repetitions.

Only qualitative analysis was carried out to observe if the bacteria has the potential to solubilize P.

270 – Table 2 … the percentage of seed germination was calculated after 10 seeds were planted?

Refer to Line 169

350 - … The result showed that IAA was synthesis … synthesized

The section has been revised

Best regards.

Round 2

Reviewer 3 Report (New Reviewer)

Review of article Effects of Sphingobium yanoikuyae SJTF8 on Rice (Oryza sativa) Seed Germination and Root Development

The manuscript was largely improved, but there is still a long way to go to completely improve article typing. There are many grammatical errors and need more writing improvements. It is highly recommended an English polishing of the manuscript to be completely revised. With some minor corrections, after English improvement, the article would be ready for acceptance. Some suggestions are provided:

69 - The genera is … is not genera (plural), Sphingobium is a genus (singular).

71 - Early studies into the genera … again.

84 – 86 - The bacterium was incorporated with a bacteria species from the Bacillus and Burkholderia to reduce the application rate of synthetic fertilizers … please rephrase.

87 - … a significantly increase in plant height which genera enabled rigorous stem wall development … rephrase.

114 - … serface … surface?

157 – The purified colony of S. yanoikuyae… scientific names in italics.

232 - A total of nine bacteria strains were isolated … bacterial.

231 – Strain identification paragraph … Why is the information of other strains added when these strains are not used in the work? How do you know that the strain is SJTF8 if you only compare 16S sequences?

235 – “The other trains with partical genomes and unspecified description … It is not understood what they want to write here.

239 - The Sphingobium genus is found under the class Alphaproteobacteria … is not found, is classified in Alphaproteobacteria.

244 – The bacterium used in this study is from the Sphingobium genera … delete sentence, unnecessary. Also, is a genus not genera (plural).

245 – 246 - The genus consists of several isolates that are well known … isolates?, or want to say species?

263 - Figure 2. Hemolysis test of S. yanoikuyae SJTF8 on the blood agar … I think it is intended to show that there is no reaction, so add that it is negative.

265 - P is the second most significant element after nitrogen (N) … begin a sentence with the full word Phosphorus.

313 - … into healtry plants … into healthy plants.

316 – … by Spaepen [46] … is Spaepen and Vanderleyden, again in 403.

334 - When a specie of S. yanoikuyae was combined isolated in Taiwan and comnibed … It is not understood what they want to write here.

337 - … fertilizer application in the crops production by up to 50 percent … reference?

365 - … are in the dorment … dormant.

379 - (Brassica campestris ssp pekinensis) … write ssp. And normal, only the name must be in italics.

408 – 409 - Figure 7. The concentration of indole-3 acetic acid (IAA) by associated S. yanoikuyae SJTF8 bacteria … add error bars to the graph.

Author Response

Greeting,

Thank you for your comments and suggestions. Please find below our response to your queries.

Note that we have added a few more citations in the Introduction and Results & Discussion sections. The captions of Figures 2-7 were revised and the manuscript layout was rearranged.

The current revised manuscript (attached) will be submitted to the MDPI English reviewing platform to correct any grammar issues.

No.

Comments and Suggestions for Authors

Response

1

69 - The genera is … is not genera (plural), Sphingobium is a genus (singular).

Word edited as mentioned (Line 69).

2

71 - Early studies into the genera … again.

Word edited as mentioned (Line 71).

3

84 – 86 - The bacterium was incorporated with a bacteria species from the Bacillus and Burkholderia to reduce the application rate of synthetic fertilizers … please rephrase.

Sentence revised (Lines 83-84).

4

87 - … a significantly increase in plant height which genera enabled rigorous stem wall development … rephrase.

Sentence revised (Lines 86-87).

5

114 - … serface … surface?

Word edited as mentioned (Line 114).

6

157 – The purified colony of S. yanoikuyae… scientific names in italics.

The scientific name is edited to italics (Line 156).

7

232 - A total of nine bacteria strains were isolated … bacterial.

Lines 232-233 have been revised.

8

231 – Strain identification paragraph … Why is the information of other strains added when these strains are not used in the work? How do you know that the strain is SJTF8 if you only compare 16S sequences?

To provide information on why the targeted bacteria species were selected. The sentence was revised and a citation was provided Lines 232-236).

9

235 – “The other trains with partical genomes and unspecified description … It is not understood what they want to write here.

The sentence was revised (Lines 235-236).

10

239 - The Sphingobium genus is found under the class Alphaproteobacteria … is not found, is classified in Alphaproteobacteria.

The sentence has been revised (Line 239).

11

244 – The bacterium used in this study is from the Sphingobium genera … delete sentence, unnecessary. Also, is a genus not genera (plural).

Point noted, and the sentence is omitted as advised (Lines 243-245).

12

245 – 246 - The genus consists of several isolates that are well known … isolates?, or want to say species?

Revised and the word isolates were replaced with species (Line 244).

13

263 - Figure 2. Hemolysis test of S. yanoikuyae SJTF8 on the blood agar … I think it is intended to show that there is no reaction, so add that it is negative.

The caption of Figure 2 was revised (Lines 261-262).

14

265 - P is the second most significant element after nitrogen (N) … begin a sentence with the full word Phosphorus.

Edit in the full word instead of the abbreviation (Line 264).

15

313 - … into healtry plants … into healthy plants.

Edit spelling error (Line 313).

16

316 – … by Spaepen [46] … is Spaepen and Vanderleyden, again in 403.

The citation has been corrected (Lines 316, 409-410)

17

334 - When a specie of S. yanoikuyae was combined isolated in Taiwan and comnibed … It is not understood what they want to write here.

The section was revised to clarify (Lines 333-337)

18

337 - … fertilizer application in the crops production by up to 50 percent … reference?

The sentence was revised, and citations were included (Line 337).

19

365 - … are in the dorment … dormant.

Edited accordingly (Line 365)

20

379 - (Brassica campestris ssp pekinensis) … write ssp. And normal, only the name must be in italics.

Edited accordingly (Line 397)

21

408 – 409 - Figure 7. The concentration of indole-3 acetic acid (IAA) by associated S. yanoikuyae SJTF8 bacteria … add error bars to the graph.

Error bars are inserted (Lines 416-417)

Kind regards

This manuscript is a resubmission of an earlier submission. The following is a list of the peer review reports and author responses from that submission.

Round 1

Reviewer 1 Report

The manuscript entitled "Interaction of Sphingobium yanoikuyae on the Germination and Root Development of Rice Seedlings" presents an analysis of the doses inoculum effect of a bacterium Sphingobium yanoikuyae on rice seed germination. I found several mistakes and missing information to do a complete analysis of the information presented by the authors and from this I can not recommended their publication. These are my major comments:

1. In the introduction authors need to gave some information to the readers about the microorganism they are studying since this bacterium is not a common PGPR and not only general information about PGPR.

2. Why the authors decide to isolate putative PGPR from air?... These kind of bacteria are commonly found in rhizospheric soils

3. Lines 89-97: which data bases were used to compare the 16S sequence?

4. Lines 98-103: Hemolysis test is a common analysis to determine some virulence factors in bacteria. What was the reason to test this in a putative PGPR?... Authors mentuioned that there is important  to discard the presences of pathogenic characteristics in a putative PGPR, but there are several other test to do this (as analzed exo and endo toxins presences) and not only hemolysis test

5. Why the authors decided to analyzed the levels of IAA in the plants?. IAA production is one of the mainly trait tested for PGPR, but this is tested for the ability of the bacterium to produce the phytohormone. Levels of IAA in plants can changes in responses of several biotic and abiotic factors and must not be directly relate with the presence of a bacterium inoculum

6. Mistakes are made in statistical intepretations: a significant statistical value refered to the probabilistic value must be expressed  as P < 0.05 or 0.01, and not P > 0.05

7. Statistical analysis: The authors mentioned they did a ANOVA test to analyze difference between treatments. However germination is expressed in percentage values which does not meet the assumptions of normality and  can not be analyzed by an ANOVA test. These type of data needs to be analyzed by a Chi-square or a logistic regression 

8. Table 2. all the treatments have the same standard error?

Author Response

Greeting,

Thank you for your feedback on the manuscript.

Please find attached the response/action committed to your feedback.

Kind regards

Reviewer 2 Report

1.      In Abstract section =

a.      Please explain briefly the reason why you use Sphingobium yanoikuyae compared to the others rhizobacteria.

2.      In Introduction section =

a.      Please add more information about the kinds of Rhizobacteria that have been used as bio-fertilizer.

b.      Please describe previous studies that have been done related to the plant growth promoting rhizobacteria (PGPR) and the effect to the germination and root development in rice seedlings.

c.      Please explain the characteristics of Sphingobium yanoikuyae

3.      In Materials and Methods section =

a.      It is better if you separate the section “2.7 Bacterial Concentration” become 2 section “2.7 Bacterial Concentration” and “2.8 Seed Germination”

b.      In line 153, I think you have spelling error “he samples were incubated again for another”, change become “the samples ……..”

4.      In Result and Discussion section =

a.      Please add more explanation about the result in Hemolysis and Phosphate Production.

b.      It will be better if you add microscopic photo specifically Sphingobium yanoikuyae, so we can observe the physical characteristic of Sphingobium yanoikuyae.

5.      In Conclusion section =

a.      Please add brief information related to the application of Sphingobium yanoikuyae in the real rice cultivation.

Author Response

Greeting,

Thank you for your feedback on the manuscript.

Please find below the response/action committed to your feedback and a copy of the manuscript.

  1. In Abstract section =
  2. Please explain briefly the reason why you use Sphingobium yanoikuyae compared to the others rhizobacteria.

According to literature, the bacteria strain has both medical and agricultural importance. According to Reddy (2014) and Yang et al. (2014), the taxa was indicated as a PGPR owing to its ability to trigger growth promotion in monocot plants such as rice.

It was revised accordingly.

  1. In the Introduction section =
  2. Please add more information about the kinds of Rhizobacteria that have been used as bio-fertilizer.

       Refer to Lines 50-69.

  1. Please describe previous studies that have been done related to the plant growth promoting rhizobacteria (PGPR) and the effect to the germination and root development in rice seedlings.

Studies by Gholamalizadeh et al. (2017) and Ng et al. (2012) noted improve germination and growth promotion when rice seeds were exposed to Alcaligenes spp., Bacillus spp., Corynebacterium spp., Enterobacter spp., Pantoea spp. and Stenotrophomonas spp. in their respective studies. However, since there are numerous species the extent to which they effect Rice growth promotion can be achieved by beneficial bacteria but the exact mechanism by which they induce their effects is not clearly understood.

  1. Please explain the characteristics of Sphingobium yanoikuyae

Refer to Lines 70-94.

  1. In Materials and Methods section =
  2. It is better if you separate the section “2.7 Bacterial Concentration” become 2 section “2.7 Bacterial Concentration” and “2.8 Seed Germination”

Revised accordingly, refer to Lines 150 and 159.

  1. In line 153, I think you have spelling error “he samples were incubated again for another”, change become “the samples ……..”

Revised accordingly, refer to Lines 189.

  1. In Result and Discussion section =
  2. Please add more explanation about the result in Hemolysis and Phosphate Production.

Revised accordingly

  1. It will be better if you add microscopic photo specifically Sphingobium yanoikuyae, so we can observe the physical characteristic of Sphingobium yanoikuyae.

Revised accordingly, refer to Lines 224-233.

  1. In Conclusion section =
  2. Please add brief information related to the application of Sphingobium yanoikuyaein the real rice cultivation.

Revised accordingly, refer to Lines 401-413.

Reviewer 3 Report

The manuscript entitled: “Interaction of Sphingobium yanoikuyae on the Germination and Root Development of Rice Seedlings”. This investigation look forward to determine the appropriate concentration of the bacteria S. yanoikuyae which can increase the efficiency of germination and seedling establishment of rice plants. In may opinion, The results obtained from this study should be described in a mainly physiological context that better clarifies the importance of the bacterial-root symbiosis for the development of rice plants. The way in which the work is presented now shows a series of results, but many of them are unconnected, which does not allow relevant conclusions to be drawn.

Despite this, some editing suggestion are included:

Abstract:

Line 23 : replace “The study was conducted with 6 treatments of the concentration of S. yanoikuyae bacteria, including” by “The study was conducted with six treatments of S. yanoikuyae in concentrations;”.

Line 28-31: Both paragraph describing some results of IAA are confusing. Please, rephrase.

Introduction:

Line 42-44: I suggest merge both paragraphs into one. For example: Some of the factors that cause land degradation are land use conversión [4] and intensive use of chemical fertilizers in agricultural land, which has been widely used in the world [1, 5].

Line 46: Delete “and” and replace by “resulting in”.

Line 48: You should explain some details because sustainable agriculture is very important…

Line 52: Replace “creating” by “inducing”.

Line 60: Delete: “and plant advancement”.

Line 60: Replace “The ways” by “the way”.

Line 61-67: Please specify which would be the indirect mechanisms.

Materials and methods

Line 116: Replace “25 oC” by “25°C”.

Line 129: Replace “an” by “one”.

Line 159: In my opinión, it is not necesary to include the IAA standard curve…

Results and Discussion:

Line 192: Could the authors explain why is important to evaluate hemolysis in this bacteria, which potential role is colinize roots plants.

Line 217: delete “growth after germination”.

Line 227: Replace “notable” by “significant”.

Line 257: change “The” by “the”.

Line 259: Indicates the figure of that result.

Line 270: Is it possible to quantify the number of root hairs on a given surface?

Line 314: Is there a significant difference in IAA due to the effect of inoculation?

Author Response

Greetings,

Thank you for your feedback.

Please find attached a revised copy of the manuscript and our response/action to your feedback below.

You will note that the title of the manuscript had been revised. We will write to the main editor to advise on the title and make amendments.

Best regards,

Abstract:

Line 23 : replace “The study was conducted with 6 treatments of the concentration of S. yanoikuyae bacteria, including” by “The study was conducted with six treatments of S. yanoikuyae in concentrations;”.

Revised accordingly.

Line 28-31: Both paragraph describing some results of IAA are confusing. Please, rephrase.

Revised accordingly.

Introduction:

Line 42-44: I suggest merge both paragraphs into one. For example: Some of the factors that cause land degradation are land use conversión [4] and intensive use of chemical fertilizers in agricultural land, which has been widely used in the world [1, 5].

Revised accordingly.

Line 46: Delete “and” and replace by “resulting in”.

Revised accordingly.

Line 48: You should explain some details because sustainable agriculture is very important…

Revised accordingly.

Line 52: Replace “creating” by “inducing”.

Revised accordingly.

Line 60: Delete: “and plant advancement”.

Revised accordingly.

Line 60: Replace “The ways” by “the way”.

Revised accordingly.

Line 61-67: Please specify which would be the indirect mechanisms.

 Revised accordingly.

Materials and methods

Line 116: Replace “25 oC” by “25°C”.

Revised accordingly.

Line 129: Replace “an” by “one”.

Revised accordingly.

Line 159: In my opinión, it is not necesary to include the IAA standard curve…

Results and Discussion:

Line 192: Could the authors explain why is important to evaluate hemolysis in this bacteria, which potential role is colinize roots plants.

Revised accordingly. Refer to 193-199.

Line 217: delete “growth after germination”.

Revised accordingly.

Line 227: Replace “notable” by “significant”.

Revised accordingly.

Line 257: change “The” by “the”.

Revised accordingly.

Line 259: Indicates the figure of that result.

Revised accordingly.

Line 270: Is it possible to quantify the number of root hairs on a given surface?

Considered for future studies.

Line 314: Is there a significant difference in IAA due to the effect of inoculation?

Data output not enough for data analysis.

Reviewer 4 Report

Line 77 please remove the quotation mark from “Kaohsiung"

In table 2, figure 4, and figure 5, the authors used an “a” simple for the lower number instead of the higher value, which is mostly used in manuscripts.

Author Response

Greetings,

Thank you for your feedback.

Please find attached a revised copy of the manuscript and our response/action to your feedback below.

You will note that the title of the manuscript had been revised. We will write to the main editor to advise on the title and make amendments.

Best regards.

Comments and Suggestions for Authors

Line 77 please remove the quotation mark from “Kaohsiung"

Revised accordingly.

In table 2, figure 4, and figure 5, the authors used an “a” simple for the lower number instead of the higher value, which is mostly used in manuscripts.

It was revised accordingly.

Round 2

Reviewer 3 Report

Unfortunately, throughout the manuscript it was never clear why it is necessary to perform a hemolysis test on a bacterium that potentially improves the performance of plant roots. Regarding its ability to solubilize phosphorous, its importance should be explained more clearly from an agronomic perspective.  The lack of statistical analysis of the IAA results does not allow us to establish with certainty which treatment is more effective in the production of this compound. The conclusion of the study fails to establish a precise concentration of CFU/ml for improve rice performance at initial stages, which generates uncertainty.